# Fungal Enzyme l-Lysine α-Oxidase Affects the Amino Acid Metabolism in the Brain and Decreases the Polyamine Level

**DOI:** 10.3390/ph13110398

**Published:** 2020-11-17

**Authors:** Elena V. Lukasheva, Marina G. Makletsova, Alexander N. Lukashev, Gulalek Babayeva, Anna Yu. Arinbasarova, Alexander G. Medentsev

**Affiliations:** 1Department of Biochemistry, Peoples’ Friendship University of Russia (RUDN University), 6 Miklukho-Maklaya St., Moscow 117198, Russia; elukasheva@yandex.ru; 2Department of Biology and General Pathology, Don State Technical University, Gagarin Square 1, Rostov-on-Don 344011, Russia; mgm52@bk.ru; 3Martsinovsky Institute of Medical Parasitology, Tropical and Vector Borne Diseases, I.M. Sechenov First Moscow State Medical University (Sechenov University), 20 M. Pirogovskaya str., Moscow 119435, Russia; alexander_lukashev@hotmail.com; 4G.K. Skryabin Institute of Biochemistry and Physiology of Microorganisms, Russian Academy of Sciences, PSCBR RAS, 5 Pr. Nauki, Pushchino, Moscow Region 142290, Russia; aarin@rambler.ru (A.Y.A.); medentsev-ag@rambler.ru (A.G.M.)

**Keywords:** metabolism, brain, l-lysine α-oxidase, l-amino acid oxidase, polyamines

## Abstract

The fungal glycoprotein l-lysine α-oxidase (LO) catalyzes the oxidative deamination of l-lysine (l-lys). LO may be internalized in the intestine and shows antitumor, antibacterial, and antiviral effects in vivo. The main mechanisms of its effects have been shown to be depletion of the essential amino acid l-lys and action of reactive oxidative species produced by the reaction. Here, we report that LO penetrates into the brain and is retained there for up to 48 h after intravenous injection, which might be explained by specific pharmacokinetics. LO actively intervenes in amino acid metabolism in the brain. The most significant impact of LO was towards amino acids, which are directly exposed to its action (l-lys, l-orn, l-arg). In addition, the enzyme significantly affected the redistribution of amino acids directly associated with the tricarboxylic acid (TCA) cycle (l-asp and l-glu). We discovered that the depletion of l-orn, the precursor of polyamines (PA), led to a significant and long-term decrease in the concentration of polyamines, which are responsible for regulation of many processes including cell proliferation. Thus, LO may be used to reduce levels of l-lys and PA in the brain.

## 1. Introduction

The l-lysine (l-lys) is an essential amino acid (AA) for humans. It is used for energy production and the biosynthesis of numerous compounds, including proteins. However, in recent years, several new important effects of this AA in the human body have been discovered. For example, in the intestine, l-lys is a partial antagonist of serotonin receptors and prevents serotonin from entering the body. Due to this, nutritional lysine has a sedative (anxiolytic) effect at the level of amygdala and cerebellum, without affecting the heart rate. By this mechanism, food l-lys suppresses artificially induced diarrhea, which is regulated by serotonin [1,2]. In in vitro experiments, l-lys was found to inhibit aldosterone production induced by serotonin and agonists of its receptor. In volunteers, lysine supplement did not affect the normal level of aldosterone; however, it significantly reduced the level of aldosterone increased by tetracosactide and a low sodium diet. Therefore, l-lys supplementation may represent a new treatment for primary adrenal diseases in which corticosteroid hypersecretion is driven by overexpressed serotonin receptors [3].

l-lys does not directly reduce the level of serum glucose, but is used to treat diabetic patients, since it not only prevents the glycosylation of proteins, in particular hemoglobin, but also reduces the formation of reactive advanced glycation products [4]. In the brain, an excess of l-lys due to impaired metabolism contributes to the development of glutaric aciduria type I [5,6], pyridoxine-dependent epilepsy [7] and hyperlysinemia [8]. In addition, lysine is a regulator of behavior, in particular sensitivity to pain [9]. One recent achievement has been the discovery of a possible coupling of the pipecolin pathway of l-lys transformation in the brain with tryptophan metabolism and with the proteins responsible for the reception of thyroxin and triiodothyronine in the brain [10]. l-lys was shown to enhance angiogenesis and inhibit ischemia in brain [11], but in the case of tumors, the deprivation of angiogenesis is one of the ways to decrease their malignancy, thus the utility of this effect is unclear. 

The catabolism of l-lys in the brain is of particular interest, given the important and not yet fully understood effects of this AA in the body. l-Lysine α-oxidase (LO), one of the l-amino acid oxidases (LAAO), an enzyme of fungal origin catalyzing the oxidative deamination of l-lys to 2-keto-6-aminocaproic acid with the formation of hydrogen peroxide and ammonia, is being studied as a potent antitumor drug. Due to intensive metabolism, most tumor cells are highly sensitive to essential AA deprivation and initially it was assumed that the antitumor effect of LO was based on the deprivation of tumor cells of an essential metabolite, l-lys [12]. Later, the role of the oxidative pathway in antitumor activity was reported [13]. The enzyme mainly cleaves l-lys; however, it can also act on its structural analogues. The conversion rates of l-ornithine (l-orn) and l-arginine (l-arg) are 8.3%, and 5.8% of the l-lys rate, respectively [14]. l-Orn is a precursor of the polyamines (PA) putrescine, spermine, and spermidine, which show diverse regulatory effects in the body. Recently, it was shown that LO can be internalized in the gastrointestinal tract [15]. As LO cleaves aliphatic positively charged AAs and can penetrate cells, we reasoned that it might be able to affect the metabolism of AAs and derive PA in the brain. We therefore sought to investigate this in the current study.

## 2. Results

### 2.1. LO Pharmacokinetics: Enzyme Stays in the Brain Much Longer than in the Bloodstream

When studying the pharmacokinetic parameters of protein substances, the question of their biological activity in the bloodstream is very important, since even partial denaturation of molecules can lead to the loss of activity. In order to assess the suitability of the developed enzyme immunoassay for the determination of catalytically active LO, the direct determination of LO activity in blood serum was carried out. The enzymatic activity and LO content measured by the enzyme immunoassay in the serum of mice after a single intravenous (*i.v*.) injection at a dose of 150 U/kg are shown in Figure 1a. The dynamics of LO concentration in the blood determined by enzyme immunoassay practically coincides with the dynamics of enzymatic activity, therefore, we can conclude that enzyme immunoassay is usable to measure functionally active LO. Good correspondence of pharmacokinetic parameters determined by direct enzymatic activity measurement and enzyme immunoassay was also shown for *Escherichia coli (E. coli)*
l-asparaginase, which is used as a therapeutic [16]. Moreover, in comparison with measuring LO activity, the enzyme immunoassay is more sensitive and makes it possible to determine low protein concentrations with higher specificity. These results allowed further use of solely the enzyme immunoassay.

As the study aimed to investigate the LO effect on the metabolism of AA and PA in the brain, we determined the pharmacokinetics of this enzyme in the brain by enzyme immunoassay (Figure 1b). Intravenously injected LO quite quickly appeared in the brain of mice; after 15 min, its concentration was 87.2 ± 6.1 ng/g of tissue. As the total dose of LO was 1.5 mg/kg, it was possible to calculate that approximately 0.15% of administered LO was found in the brain.

### 2.2. LO Showed Resistance to Proteolysis

It can be assumed that the long-term preservation of LO in the brain is associated with the absence of degrading enzymes, as well as the possible high resistance of LO to proteolysis. To confirm this assumption, LO was incubated at 37 °C in the presence of the two most prominent proteolytic enzymes, trypsin and chymotrypsin, under optimal conditions for their action. It turned out that these proteinases, which have an affinity for peptide bonds formed by different AAs, did not cleave LO to reduce its activity (Figure 2).

### 2.3. Concentrations of Target Amino Acids Are Decreased in the Brain

LO catalyzes the oxidative deamination of l-lys and its structural analogues, albeit at a much lower rate. Accordingly, the concentration of these LO substrates in the brain dropped, although to a much smaller extent (Figure 3).

### 2.4. Concentrations of Amino Acids, Which Are Directly Associated with the Tricarboxylic Acid Cycle (l-asp and l-glu), Are Significantly Changed in the Brain

In addition, the enzyme significantly affected the redistribution of AAs directly associated with the tricarboxylic acid (TCA) cycle (l-asp and l-glu). The study of the dynamics of other l-AA concentrations: (valine, glycine, alanine, proline, tryptophan, isoleucine, methionine, asparagine, glutamine, phenylalanine, tyrosine) in the brain after LO *i.v.* injection did not reveal significant changes in most cases, except for l-glu, l-asp (Figure 4), and l-val.

### 2.5. LO Caused a Significant and Long-Term Decrease of the Polyamines Concentrations 

As LO degrades l-orn, a precursor of PA, it was of interest to investigate the possibility of using this enzyme to affect the content of PA in the brain. LO significantly reduced the concentration of PA in the brain, most strongly for putrescine (more than 34.2 ± 4.6% in 1 h), less for spermine, and even less for spermidine. It should be noted that, likely due to the long-term presence of LO in the brain, the effect on the concentration of putrescine (41.3 ± 3.4%) and spermine (67.2 ± 7.8%) persisted for at least 24 h (Figure 5).

## 3. Discussion

The development of new drugs is an eternally relevant task. Despite many achievements, the spectrum of available antitumor drugs is not exhaustive. LO was previously characterized as an enzyme endowed with cytotoxic activity. To date, the main mechanisms of its action have been established—the depletion of the essential amino acid l-lys and cell death promoted by hydrogen peroxide produced as a reaction by-product [12,13]. The study of antitumor activity showed that LO is effective in the treatment of a wide range of malignant diseases [12]. When studying the pharmacokinetics of LO in tissues, we found that the enzyme is specifically accumulated in the brain at levels not expected for a large protein. As the total dose of LO for pharmacokinetic study in mice was 1.5 mg/kg, we calculated that approximately 0.15% of administered LO was found in the brain. LO was retained in the brain for a significant time, as indicated by the mismatch of the LO pharmacokinetics in the brain and blood (Figure 1a). Intravenously administered LO was rapidly eliminated from the bloodstream (Figure 1a). We showed that LO is resistant to the action of proteinases, which cleave different kinds of bounds: trypsin specifically hydrolyzes peptide bounds at the carboxyl group of lysine and arginine residues, chymotrypsin predominantly breaks down peptide bounds after the residues of aromatic amino acids (Figure 2). The isoelectric point of LO is 4.25, which is the reason for the LO negative charge at blood at pH 7.4 [12]. Negative charge is caused by the presence of glutamate or aspartate amino acid residues. The peptide bounds formed by these amino acids are difficult to hydrolyze with chymotrypsin and trypsin as well as by other serum proteinases. It can be assumed that, like many other proteins, LO gets into cells by endocytosis. Endocytic vesicles merge with lysosomes, where the pH is significantly lower than in serum, which leads to a decrease in the negative LO charge. Low pH values are optimal for protein hydrolysis by lysosomal enzymes. T_1/2_ of LO in serum was 1.23 ± 0.10 h. Most of the antitumor enzymes that are injected *i.v.* have similar T_1/2_ in blood, for example, T_1/2_ of methionine gamma-lyase from *C. tetani* was 1.71±0.14 and l-asparaginase from *E.coli* about 3 h; for l-asparaginase from *E. coli* it was shown that human macrophages bind and degrade this enzyme [16,17,18,19]. While the value of LO T_1/2_ in blood is rather short, T_1/2_ of LO in the brain was 9.41 ± 1.10 h. The longer persistence in the brain is possibly due to rather slow utilization of extracellular proteins in the brain. This differential pharmacokinetics is compatible with penetration of the blood–brain barrier, rather than detection of the residual enzyme in blood vessel of the brain. To date, only few cases of proteins penetrating the BBB are known. For instance, lactoferrin (M.M. 80 kDa), which is close to the size of the LO subunit (60 kDa), penetrates into the brain, likely using corresponding receptors for the transfer of this protein [20]. It then affects cell differentiation and is considered as an alternative drug for the chemotherapy of tumors [21,22]. Moreover, there are a number of publications on the possibility of lactoferrin or its conjugated forms being used as vehicles in a system for delivering certain genes to the brain for gene therapy [23]. Some proteins also enter the brain, for example, regulatory proteins of the body such as ghrelin, leptin, and some others, including prion proteins, contributing to their ability to induce disease in the brain [24].

The long-term LO resistance to proteolysis and presence in active form in the brain is also illustrated by the dynamics of l-lys concentrations after LO administration (Figure 3). Indeed, the l-lys content in the brain is decreased by 20% 15 min after LO injection, and even 6 h later—by up to 60%. 

LO oxidatively deaminates l-lys and its structural analogues, l-arg (5.9% of activity towards l-lys) and l-orn (5.3% activity). It is noteworthy that the level of l-arg and l-orn in the brain was minimal 1 h after the LO injection, and then it gradually began to return to its original values (Figure 3). The drop in the l-lys concentration by 6 h had reached a minimum of 35.2 ± 6.5% of the initial. The decrease was stronger and lasted much longer, than the decrease in l-arg and l-orn. The concentration of l-arg and l-orn, as well as l-lys, did not return to the initial values even 24 h after the LO injection (49.4 ± 8.5% of the initial), suggesting the preservation of active LO in brain. The difference in the AA concentration dynamics can be explained by the fact that l-lys is an essential AA, and its concentration in the brain cannot be replenished by metabolic pathways, but only from food sources or due to the breakdown of body proteins. The content of l-arg and l-orn returned to normal values faster since they can be synthesized in the body. Even 24 h after LO administration, the l-lys concentration did not reach 50% of the initial level. This indicates that all this time the cells experienced lysine starvation. In mammals, l-Lys undergoes catabolism mainly through the saccharopine pathway [25]. However, a direct oxidation pathway of the alpha-amino group of l-lys was discovered in the brain, followed by the formation of Δ1-piperideine-2-carboxylate and pipecolic acid [26,27]. To date, the presence of both pathways of l-Lys catabolism in the brain have been confirmed: the saccharopine pathway in mitochondria, which is less active than the pipecolic acid pathway in peroxisomes. The latter pathway has been found to be more specific for the brain [6]. Under the influence of fungal LO, l-lys catabolism flows through l-pipecolate, which has been shown to be responsible for the cytotoxic properties of LOs [28]. Therefore, the interference of LO in metabolic processes can be promising for diseases associated with elevated levels of l-lys and saccharopine [29].

The decrease in the l-glu concentration lasted as long as that of l-lys. l-glu is used for the biosynthesis of l-orn and l-arg. As these AAs are partially destroyed by LO, it is possible that the drop in the l-glu concentration resulted from the effect of LO on l-orn and l-arg. Since LO shows antitumor effects in the body [12,30], it can be assumed that the decrease of l-glu, which is critical for the proliferation of many tumor strains [31], may be one of the mechanisms of LO antitumor action along with l-lys depletion and the oxidative pathway [13]. It is rather difficult to explain the more than 20% decrease in the l-val concentration 1 h after *i.v*. LO administration since the metabolism of AAs with branched chains is not directly associated with the metabolism of l-lys. Unexpectedly, there was a significant long-term increase of the l-asp concentration in the brain. The content of this AA was about 140% 24 h after *i.v.* LO injection. We can assume the implementation of the following biochemical changes in the metabolic pathways. An intermediate product of l-lys catabolism is glutaric acid, which has been shown to be an inhibitor of glutamate dehydrogenase [32]. It may slow down the conversion of decreased amounts of l-glu (Figure 4) to α-ketoglutarate, which is the substrate in transamination reaction of l-asp conversion to oxaloacetate, a participant in the TCA cycle.

Since it was known that LO degrades l-orn, a precursor of PA, it was of interest to investigate the possibility of using this enzyme to affect the content of PA in the brain. PA are low molecular weight organic compounds containing two or more amino groups. In the nervous tissue, three main PA are present: spermine, spermidine, and putrescine. Both in the body as a whole and in the CNS, PA are important regulators in protein and nucleic acid biosynthesis and play role in the stabilization of membranes and modulation of neuronal activity. PA can mediate the activation of protein biosynthesis; therefore, the inhibition of this process leads to restriction in cell growth. PA play a particularly important role in the processes of neuronal cell division, differentiation, axon and synaptogenesis, and in the mechanisms of synaptic plasticity. Intracellular polyamines or those coming from outside stimulate rapid cell proliferation, and depletion of cellular PA leads to the inhibition of proliferative processes [33,34]. Various methods of slowing down PA biosynthesis are currently being studied. For example, ornithine decarboxylase inhibitors affect PA concentrations in tumors, which is considered a promising approach to tumor suppression [35]. LO significantly reduced the concentration of PA in the brain, most strongly for putrescine (more than 34.2 ± 4.6% in 1 h), less for spermine and even less for spermidine. It should be noted that, likely due to the long-term presence of LO in the brain, the effect on the concentration of putrescine (41.3 ± 3.4%) and spermine (67.2 ± 7.8%) persisted for at least 24 h (Figure 5). The decrease of PA level under the influence of LO can be proposed as a new mechanism of LO antitumor action.

Comparison of Figure 3 and Figure 5 shows that the drop in the concentration of l-orn and PA is not directly related. l-arg, along with l-orn, is a molecule from which PA can be synthesized through agmatine or after transformation into l-orn. PA homeostasis in tissue is achieved by a careful balance between synthesis, degradation, and uptake. The first and what is often considered the rate-limiting step in the biosynthesis of the PA is catalyzed by the enzyme ornithine decarboxylase. Mammalian ornithine decarboxylase has a fast turnover, with a half-life as short as a few minutes, so its cellular level is thus rapidly changed when the rate of PA synthesis or degradation is changed [36,37]. It has been established that ornithine decarboxylase is inhibited by putrescin [38]. An initial drop in putrescine concentration can activate ornithine decarboxylase and cause an even greater drop in PA concentrations (Figure 5). A total of 24 h after LO introduction, the levels of nonessential amino acids l-orn (95.4 ± 7.4%) and l-arg (76.1 ± 8.5%) gradually move up to normal values, as well as the concentration of spermidine (Figure 5). At the same time, concentrations of other PA remain significantly lower: spermine—67.2 ± 7.8%; putrescine—41.3 ± 3.4% of the initial. PA concentrations in the putrescine–spermidine–spermine sequence are not linearly related but are the result of complex processes. It has been established that enzymes for the synthesis and decomposition of PA are present in neurons, but absent in glial cells. However, after synthesis PA are transported by carriers into glial cells, where they are accumulated at higher concentrations than in neurons [39,40]. Given the higher content of PA in glia and quantitative predominance of glial cells over neurons, it may be assumed that in our experiments with brain homogenates we determined the total PA amount, which reflects not only the process of PA synthesis and decomposition, but, as well the transfer and accumulation of PA in glia. Direct relationship between conversion of l-orn and l-arg to putrescine and then putrescine to spermidine and from spermidine to spermine and the decomposition of PA cannot be detected in experiments on a general homogenate of brain tissue.

## 4. Materials and Methods

### 4.1. Materials

O-Dianisidine dihydrochloride, 3,3′,5,5′-tetramethylbenzidine (TMB), phosphate buffered saline (PBS), goat anti-rabbit IgG-peroxidase antibody, putrescine, spermidine, spermine, 1,7-diamino heptane, benzoyl chloride, sulfuric acid, phosphoric acid, ethanol, methanol, l-lysine monohydrochloride, and other amino acids were purchased from Sigma-Aldrich; horseradish peroxidase from Bio-Rad California (Berkeley, CA, USA), Coomassie brilliant blue G-250, Na_2_HPO_4_, NaH_2_PO_4_, Na_2_CO_3_, NaHCO_3_ from Merck (Darmstadt, Germany), Freund’s complete adjuvant and Freund’s incomplete adjuvant from Difco Laboratories (Franklin Lakes, NJ, USA), 0.9% NaCl solution, oxymethyl aminomethane from PanEco (Moscow, Russia), trypsin from BDH Chemicals (London, UK), chymotrypsin from Fluka (St. Louis, MO, USA); HCl, CaCl_2_, and chloroform from Chemmed (Moscow, Russia).

### 4.2. Animals

SPF mice of the Balb/c line with an average body weight 20 g were obtained from the breeding facility of the M. M. Shemyakin and Y. A. Ovchinnikov Institute of Bioorganic Chemistry (Pushchino, Russia). Each experimental group included 16 mice. The experiments were approved by the local ethics committee of Peoples’ Friendship University of Russia, with a decision on 17/09/2015. Animals were euthanized by diethyl ether anesthesia followed by cervical dislocation. For blood collection, diethyl ether anesthesia was used.

Chinchilla rabbits (2 female) up to 3 kg weight, 3–4 months old were used to produce antibodies against LO. All animals were kept in the vivarium of the Scientific Research Institute of Mitoengineering (Moscow, Russia).

### 4.3. l-Lysine α-Oxidase

LO (M.M. 120 kDa, consisting of two identical subunits) was purified according to a published method [41] from a culture of *Trichoderma* cf. *aureoviride* Rifai VKM F-4268D and packed into 10 mL sterile flasks (10 mg/flask). Immediately before administration, LO was dissolved in 1 mL of sterile water and diluted with 0.9% sterile sodium chloride solution to concentration 0.1 or 0.2 mg/mL. LO was administered intravenously (*i.v*.) at a concentration of 0.1 mg/mL (for doses 1.0 and 1.5 mg/kg) or 0.2 mg/mL (for dose 3.0 mg/kg). The injected volume was 160–340 µL.

### 4.4. Blood Sampling

Samples (50 µL) of blood were taken from the retroorbital sinus (intervals up to 3 h) or from the jugular vein (100 µL) after killing mice by cervical dislocation (9 and 24 h). After LO injection, blood was taken from eight animals after 10 min, 1, 6, and 24 h and from another eight animals after 30 min, 3 and 48 h. Blood sampling time was calculated individually for each animal in accordance with the time of LO administration. Within 1 h of sampling, the blood was centrifuged (3000 rpm, 1 min) and serum was collected.

### 4.5. Primary Antibodies

Antibodies to LO were obtained after double immunization of two rabbits. For each immunization, 2 mg of LO in 1 mL 0.02 M phosphate buffer pH 7.4 were used, with the addition of 1 mL of complete (at the first immunization) or incomplete (at the second immunization) Freund’s adjuvant. The enzyme was administered intradermally. The second immunization was performed 30 days after the first one. Blood was taken every 3 days in the period from 10 to 20 days after the second injection. According to the preliminary titration, serum with the maximum titer of antibodies to LO was used for the study. 

### 4.6. LO Determination in Blood Plasma by Enzyme Immunoassay

For this assay, 96-well plates were coated with pure LO antigen. The test serum samples were applied to wells, then immune serum was introduced. The unbound antibodies and liquid-phase LO-antibody complexes were washed away, and the plates were processed with secondary antibodies to the rabbit immunoglobulins conjugated with horseradish peroxidase, then washed, and TMB solution was added. The optical density in the wells after staining was determined spectrophotometrically at a wavelength of 450 nm. The maximum signal was detected without sample addition (zero level). If necessary, samples were additionally diluted to fit the calibration curve.

### 4.7. LO Enzymatic Activity 

LO activity was measured by the rate of H_2_O_2_ formation in 20 mM Tris-phosphate buffer (pH 7.8) in the presence of o-dianisidine (0.2 mM), peroxidase (5 μg/mL), and l-lys (2.0 mM) on a Hitachi (U-2900) spectrophotometer (E_436_ = 8.3 mM^−1^cm^−1^). One unit of activity (U) was defined as the amount of an enzyme that catalyzes the oxidation of 1 μmol of l-lys per minute at 25 °C.

### 4.8. Protein

The amount of protein in samples was determined by the Bradford method [42].

### 4.9. Polyamines (PA)

PA levels in the brain were determined after converting them to benzoyl derivatives by high-performance liquid chromatography [43,44]. 1,7-Diamino heptane was used as internal control. To precipitate the proteins, 2 mL HClO_4_ (1.0 M) was added to 0.2 g tissue. After centrifugation, 2.0 mL NaOH (2.0 M) was added to 1.0 mL of supernatant for neutralization. Right before use (not longer than 1 min) benzoyl chloride (10 μL) was mixed with methanol (10 μL) and added to the mixture. After intensive stirring for 2 h, 1.0 mL of chloroform was added, and the sample was centrifuged for 10 min at 350× *g*. The chloroform fraction was removed, 1.0 mL of chloroform was re-added to the samples, and samples were centrifuged. After that, the chloroform fractions were evaporated at 80 °C. The dry residue was dissolved in 100 µL of 60% methanol. The composition of the mobile phase was methanol:water (60:40), the volume of the samples injected to Lichrospher RP18 column was 20 µL and the chromatographic analysis time was 45 min. Substances were identified by absorption at 229 nm. The exit time from a chromatographic column for the benzoyl derivatives were putrescine 12 min, spermidine 19 min, 1,7-diamino heptane 21 min, and spermine 30 min.

### 4.10. Resistance of LO to Proteolysis

The resistance of LO to proteolysis was investigated using two proteinases with an affinity for peptide bonds formed by various AAs. LO solution (0.17 μM) was incubated at 37 °C in 0.2 M Tris-HCl buffer (pH 7.8), containing 0.02 M CaCl_2_, for 2 h. Trypsin and chymotrypsin concentrations in the media were significantly higher than the LO concentration, i.e., 21.7 and 20 μM, respectively. In parallel, the control was incubated without trypsin or chymotrypsin addition. Samples were taken every 30 min to determine LO activity. 

### 4.11. Statistical Analyses

All values are expressed as mean ± standard error of the mean (SEM). Student’s *t*-test was used to analyze the differences between the control and experimental groups. To study the concentration of LO in plasma and brain after administration, we used coefficient of determination (R^2^) value from nonlinear regression analysis. Multiple comparisons between groups were evaluated using one-way ANOVA and statistically significant differences were recorded at *p* < 0.01 or *p* < 0.05. Data analysis was performed using IBM SPSS statistics 21.0 software.

## 5. Conclusions

It was shown for the first time that the fungal glycoprotein LO is specifically retained in the brain and can affect metabolism in this organ. The persistence of LO in the brain for up to 24 h after *i.v.* injection is suggested by its exclusion from general metabolism and specific pharmacokinetics in the brain. 

LO actively intervenes in AA metabolism in the brain. The most significant impact of LO is towards AA that are directly exposed to its action (l-lys, l-orn, l-arg). The depletion of l-orn, the precursor of polyamines, leads to a significant and long-term decrease of their concentrations. As polyamines are responsible for many processes, including cell proliferation, the decrease of their level in tissue by LO should be further investigated as a potential additional mechanism of LO antitumor action.

## Figures and Tables

**Figure 1 pharmaceuticals-13-00398-f001:**
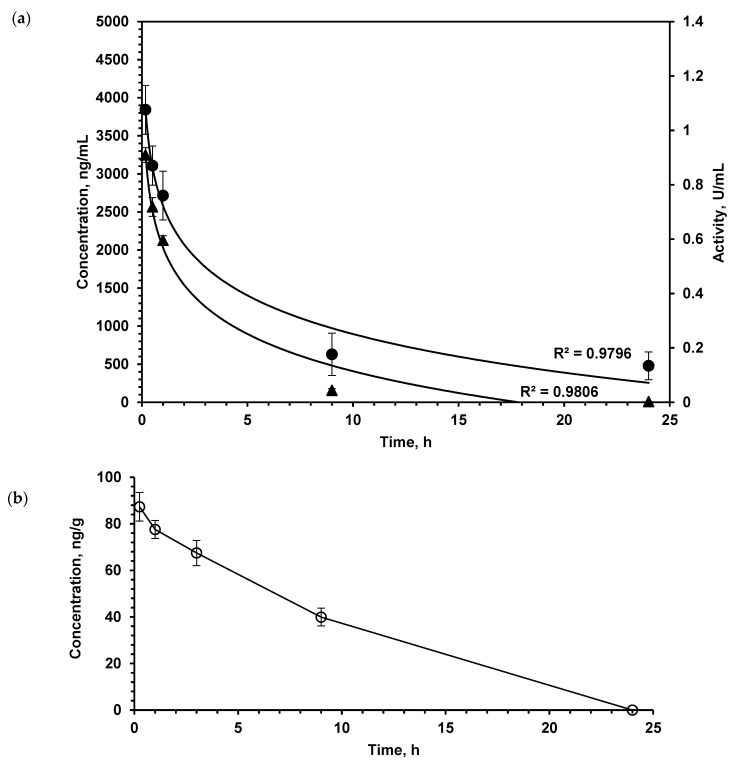
(**a**) l-lysine α-oxidase concentration (▲, left OX axis) and activity (•, right OX axis) in plasma after single *i.v.* administration at a dose of 1.5 mg/kg. (**b**) l-lysine α-oxidase concentration in the brain tissue after single *i.v.* administration at a dose of 1.5 mg/kg. Number of animals *n* = 5 at each time point. The values shown are mean of three determinations. Results are presented as mean ± SEM.

**Figure 2 pharmaceuticals-13-00398-f002:**
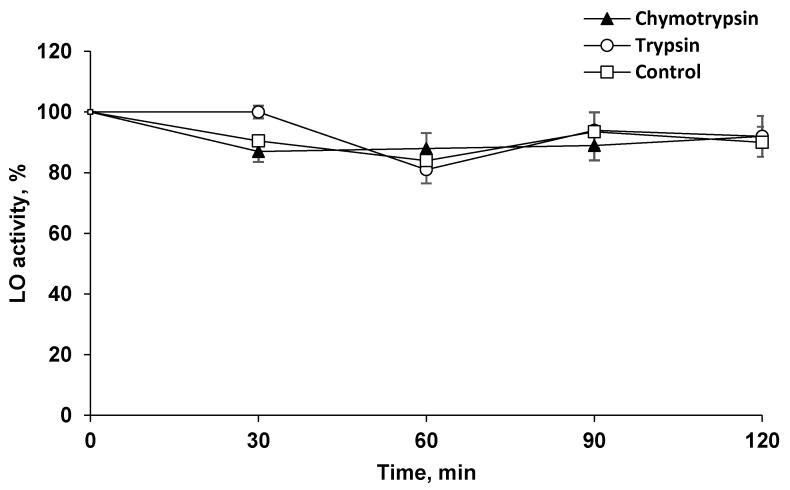
The resistance of l-lysine α-oxidase (LO) to proteolysis. LO (0.17 μM) incubation at 37 °C in 0.2 M Tris-HCl buffer (pH 7.8) in the presence of 21.7 μM chymotrypsin (▲), 20 μM trypsin (○), without additives—control (□).

**Figure 3 pharmaceuticals-13-00398-f003:**
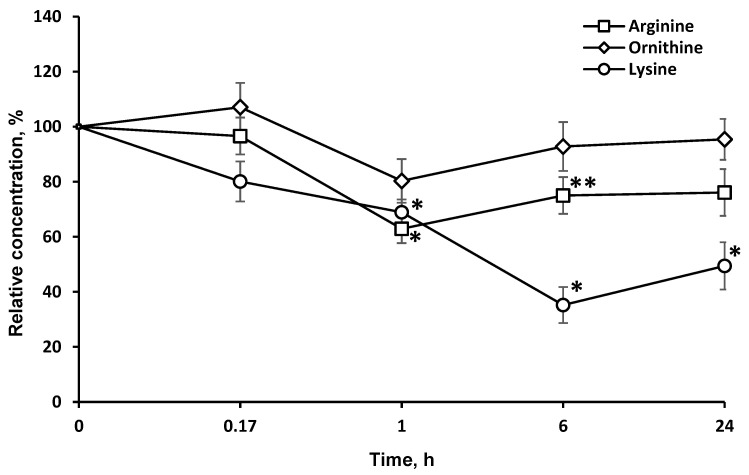
Dynamics of l-lysine structural analogs concentrations in the brain after a single l-lysine α-oxidase injection *i.v.* at a dose of 1 mg/kg. The results are presented as mean ± SEM, as a ratio to the control group. Difference between experimental and control groups was assessed using one-way ANOVA. Statistically significant difference designated as ** for *p* < 0.05 or * for *p* < 0.01.

**Figure 4 pharmaceuticals-13-00398-f004:**
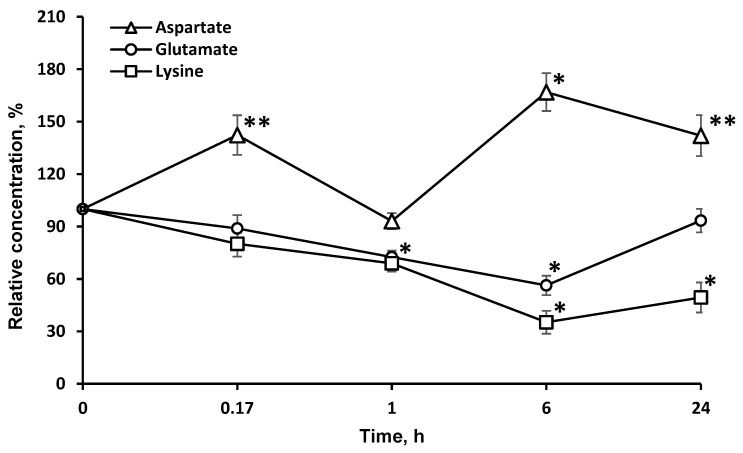
Dynamics of amino acid concentrations in the brain after a single l-lysine α-oxidase injection *i.v.* at a dose of 1 mg/kg. The results are presented as mean ± SEM, as a ratio to the control group. Difference between experimental and control groups was assessed using one-way ANOVA. Statistically significant difference designated as ** for *p* < 0.05 or * for *p* < 0.01.

**Figure 5 pharmaceuticals-13-00398-f005:**
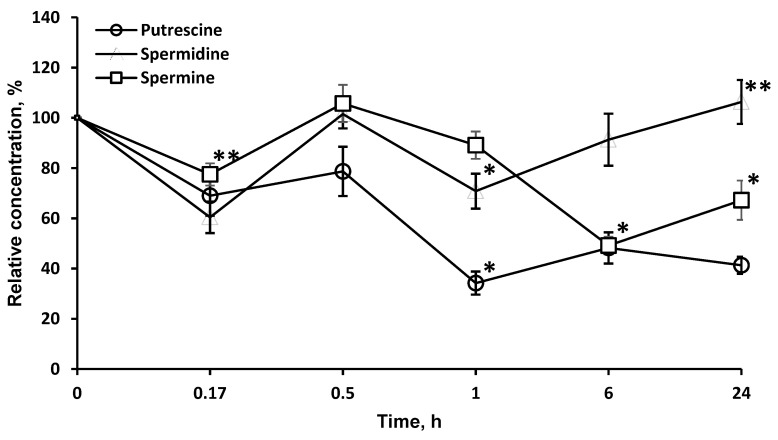
Dynamics of PA concentrations in the brain after a single l-lysine α-oxidase injection *i.v.* at a dose of 1 mg/kg. The results are presented as mean ± SEM, as a ratio to the control group. Difference between experimental and control groups was assessed using one-way ANOVA. Statistically significant difference designated as ** for *p* < 0.05 or * for *p* < 0.01.

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
