# Peer review of "Fungal Enzyme l-Lysine α-Oxidase Affects the Amino Acid Metabolism in the Brain and Decreases the Polyamine Level"

_pharmaceuticals, 2020, doi:10.3390/ph13110398_

Round 1

Reviewer 1 Report

In this manuscript Dr. Lukasheva and colleagues, report the investigation of the effects of lysine-a-oxidase on the metabolism of amino acids and biogenic polyamines in the brain. The manuscript contains interesting data and observations. However, reading of the text generated a set of remarks and questions.

Specific comments (one by one):

  • line 29 (Abstract). “Thus, LO may be used as an antitumor agent in the brain…” – the manuscript deals nothing with the activity of LO even against tumor cells. Such an ambitious conclusion must be based on detailed studies, but not on general assumptions.
  • line 60 (page 2). It must be aminocaproic, but not “aminokaproic”.
  • line 61 (page 2). “Most tumor cells are highly dependent upon essential AAs…” – normal cells are also dependent on essential amino acids…
  • line 64 (page 2). What do you mean under “The enzyme mainly cleaves L-lys (100%); however it can also…” ? The activity towards Lys was taken for 100% ? The sentence will look better if 100% will be deleted
  • Figure 1 (page 3) looks strange for me. First impression is that the amount of LO in brain is much higher than in plasma and that in brain LO is much more stable. These are because of double OX axis. It may be better either to provide two separate graphs (plasma and brain), or to use logarithmic scale, or smth else… It is also unclear what happens with LO in plasma after 9 h (24 h point looks like a zero, but only looks like...). At 9 h the concentration of LO in plasma is about 100 ng/mL, i.e. same as LO concentration in brain just after the injection. Of course, in the text there are τ½ values, but still… Besides, for the manuscript and for readers the activity of LO, but not the amount of the protein, is essential. Please provide data on the activity of the enzyme in plasma and brain. The remark from page 2 (lines 74-75) “It was previously shown that LO determined by this method is catalytically active [14]” and remark from page 5 (lines 153-154) “It was previously shown that determined by an enzyme immunoassay LO is catalytically active [23]” are not convincing, since ref.14 deals nothing with immunoassay and ref.23 is Abstract of FEBS Congress, but not a paper in a per-reviewed journal.
  • Figure 2 (page 3) provides essential and interesting data about the stability of LO towards trypsin and chymotrypsin. However, the reasons of LO degradation in brain and plasma are not disclosed and discussed, i.e. what may be possible mechanism and who are the players. The use of only two serine endopeptidases strongly limits the significance of the stability data. May be pronase will split LO ? What about exopeptidases ?
  • Lines 108-111 (page 4) and lines 179-181 (pages 6) are identical. Please eliminate the repeat.
  • Figure 5 (page 5). Up to 1 h time the dynamics of polyamine concentrations are as it may be expected. Put comes from Orn; Spd comes from Put and dcAdoMet; Spm comes from Spd and dcAdoMet. Respectively, the depletion of Orn pool must affect first Put and then Spd pool. This is typical for ornithine decarboxylase (ODC) inhibitors, most known is a-difluoromethyl ornithine (DFMO, Eflornithine®), which efficiently depletes Put and Spd pool, but is much less effective towards Spm one. However, after 24 h Spd concentration is not changed, while the concentration of Spm is about 70% and Put is of only 40% of the initial levels. In general, it is not easy to deplete Spm pool and does not change Spd concentration even in cell culture experiments. The observed dependences (Fig.5) are unusual and not discussed in the manuscript.
  • lines 171-172 (page 6). Instead of “piperidine-2-carboxylate” it must be Δ1-piperideine-2-carboxylate
  • line 185 (page 6). Why “… it can be assumed that the removal of L-glu, which is critical for the proliferation of tumor cells…” ? From Fig.4 it is clear that about 60% of Glu is left after 6 h. What do you mean under “removal” ? Besides, the centrality of Glu metabolism makes this amino acid crucial for normal cells too. Please modify the sentence.
  • lines 208-210 (page 6). “However, administration of these inhibitors is accompanied by a number of neurological side effects, which are apparently associated with the peculiarities of PA metabolism in the brain”. This is not a true statement. For example, DFMO (this drug was approved by WHO) is used for chemoprevention [Gerner E.W., et al. Biol.Chem., 2018, 293, 18770–18778] and to cure late-stage of sleeping sickness disease [Steverding D. Parasites & Vectors, 2010, 3, 15]. The same is true also for some other inhibitors, see the review [Casero R.A., et al. “Polyamine metabolism and cancer: treatments, challenges and opportunities”. Nat.Rev.Cancer, 2018, 18, 681-695].
  • line 251 (page 7). “… as the discovery…” Discovery seems too ambitious here. This is an interesting experimental fact, which still requires at least a proper discussion, but not a “discovery”.
  • line 219 (page 7) and line 154 (page 5). References [15] and [23] are essential for the manuscript and ref [15] is placed even in conclusion. Both references are Abstracts. Please substitute these with the references on the papers in per-reviewed journals.
  • line 294 (page 8). “…followed by 20 μl benzoyl chloride (diluted in methanol 1:1)”. This sounds more than strange because of the formation of methyl benzoate in such a solution. Respectively, the excess of benzoyl chloride is unpredictable. This makes the results at least hardly reproducible, or might be even not completely reliable. As a rule, benzoyl chloride is used as such for pre-column modification of polyamines [Li B., et al. ACS Chem. Biol., 2016, 11, 2782-2789], or it may be diluted with inert water-soluble organic solvents. For example, dansylation of polyamines is performed using a solution of dansyl chloride in acetone [Minocha R.& Long S. J. A 2004, 1035, 63–73].
  • Line 297 (page 8). For HPLC determination of polyamines 1,7-diaminoheptane is traditionally used as inner standard, but not “1,4-diamino heptane”. Did you use 1,4-diaminoheptane ?

It is known that the etiology of 20–30% of epithelial cancers is directly associated with inflammation. Spermine oxidase (SMOX) is a polyamine catabolic enzyme that is highly inducible by inflammatory stimuli resulting in the generation of hydrogen peroxide and acrolein-producing 3-aminopropanal; these lead to DNA damage and apoptosis [Sierra J.C., et al. Oncogene, 2020, 39, 4465–4474]. In the multiple intestinal neoplasia mouse model, treatment with SMOX inhibitor reduces enterotoxigenic Bacteroides fragilis induced colon tumorigenesis by 69% [Goodwin A.C., et al. Proc.Natl.Acad.Sci USA, 2011, 108, 15354–15359]. The same is true for gastric cancer [Chaturved R., et al. Gastroenterology, 2011, 141, 1696–1708]. Respectively, the use of hydrogen peroxide generating systems for treatment of cancer is comparatively delicate.

Author Response

Dear Editor,

Thank you very much for attention to our manuscript. It has been revised according to the reviewers’ suggestions. The list of specific modifications is provided below:

Point 1: line 29 (Abstract). “Thus, LO may be used as an antitumor agent in the brain…” – the manuscript deals nothing with the activity of LO even against tumor cells. Such an ambitious conclusion must be based on detailed studies, but not on general assumptions.

Response 1: We agree with the comment of the reviewer and corrected the sentence: «Thus, LO may be used to reduce levels of L-lys and PA in the brain».

Point 2: line 60 (page 2). It must be aminocaproic, but not “aminokaproic”.

Response 2: We are sorry about the spelling mistake. Corrected.

Point 3: line 61 (page 2). “Most tumor cells are highly dependent upon essential AAs…” – normal cells are also dependent on essential amino acids…

Response 3: As tumor cells divide more often and grow faster than normal cells, they have an increased need for growth factors, including essential amino acids. Several enzymes, and in particular L-arginase, arginine deiminase, L-methionine gamma-lyase and LO have been showing promising results in vitro and in vivo studies. [Fernandes, H. S., et al. "Amino acid deprivation using enzymes as a targeted therapy for cancer and viral infections." Expert opinion on therapeutic patents 27.3 (2017): 283-297; Pokrovsky, Vadim S., et al. "Amino acid degrading enzymes and their application in cancer therapy." Current medicinal chemistry 26.3 (2019): 446-464]

corrected to “Due to intensive metabolism, most tumor cells are highly sensitive to essential AA deprivation”

Point 4: line 64 (page 2). What do you mean under “The enzyme mainly cleaves L-lys (100%); however it can also…” ? The activity towards Lys was taken for 100% ? The sentence will look better if 100% will be deleted

Response 4: Yes, the activity towards L-lys was taken for 100%. And we agree that it will be better to delete “100%” in this sentence.

Point 5: Figure 1 (page 3) looks strange for me. First impression is that the amount of LO in brain is much higher than in plasma and that in brain LO is much more stable. These are because of double OX axis. It may be better either to provide two separate graphs (plasma and brain), or to use logarithmic scale, or smth else…

It is also unclear what happens with LO in plasma after 9 h (24 h point looks like a zero, but only looks like...).

At 9 h the concentration of LO in plasma is about 100 ng/mL, i.e. same as LO concentration in brain just after the injection. Of course, in the text there are τ½ values, but still… Besides, for the manuscript and for readers the activity of LO, but not the amount of the protein, is essential. Please provide data on the activity of the enzyme in plasma and brain. The remark from page 2 (lines 74-75) “It was previously shown that LO determined by this method is catalytically active [14]” and remark from page 5 (lines 153-154) “It was previously shown that determined by an enzyme immunoassay LO is catalytically active [23]” are not convincing, since ref.14 deals nothing with immunoassay and ref.23 is Abstract of FEBS Congress, but not a paper in a per-reviewed journal.

Response 5: We agree that using two axes (right and left) in the same figure can be confusing. Figure 1 was split into two separate graphs Fig1a and Fig.1b.

The elimination of protein from the bloodstream is related to its distribution throughout the organs. Significant breakdown of blood plasma proteins occurs mainly in the liver macorphages. In tissues, the LO half-life is longer, than in serum. Most of the enzymes that are injected intravenously have T1/2 similar to LO, for example, T1/2 of methionine gamma-lyase is not very high. T1/2 of native proteins in blood makes scientists search for the ways of protein conjugation with some neutral molecules, for instance with polyethylene glycol. This approach aims prolongation of T1/2 and immunogenicity reduction [Pasut, Gianfranco, Mauro Sergi, and Francesco M. Veronese. "Anti-cancer PEG-enzymes: 30 years old, but still a current approach." Advanced drug delivery reviews 60.1 (2008): 69-78; Zhang, Xiaopei, et al. "Encapsulating therapeutic proteins with polyzwitterions for lower macrophage nonspecific uptake and longer circulation time." ACS Applied Materials & Interfaces 9.9 (2017): 7972-7978.].

Yes, LO content in brain is lower, and we estimated: (line78-79) “As the total dose of LO was 1.5 mg/kg, it was possible to calculate that approximately 0.15% of administered LO was found in the brain”.  So, indeed, the amount of enzyme in brain is small, but LO stays in the brain longer than in serum.

Point 6: Figure 2 (page 3) provides essential and interesting data about the stability of LO towards trypsin and chymotrypsin. However, the reasons of LO degradation in brain and plasma are not disclosed and discussed, i.e. what may be possible mechanism and who are the players. The use of only two serine endopeptidases strongly limits the significance of the stability data. May be pronase will split LO ? What about exopeptidases?

Response 6: We agree that the question of LO resistance to proteolysis is of special interest. The possibility of peptide bonds cleavage in most cases depends not on the nature of amino acid residues in the enzyme active center, but on the substrate specificity of enzymes. We used these proteinases because they cleave different bonds: trypsin specifically hydrolyzes peptide bonds at the carboxyl group of lysine and arginine residues, chymotrypsin predominantly breaks down peptide bonds after the residues of aromatic amino acids. We are grateful to the reviewer for his comment and we discussed this question in article:

«We showed that LO is resistant to the action of proteinases, which cleave different kinds of bounds: trypsin specifically hydrolyzes peptide bounds at the carboxyl group of lysine and arginine residues, chymotrypsin predominantly breaks down peptide bounds after the residues of aromatic amino acids (Fig. 2). The isoelectric point of LO is 4.25, which is the reason for the LO negative charge at blood at pH 7.4 [12]. Negative charge is caused by the presence of glutamate or aspartate amino acid residues. The peptide bounds formed by these amino acids are difficult to hydrolyze with chymotrypsin and trypsin as well as by other serum proteinases. It can be assumed that, like many other proteins, LO gets into cells by endocytosis. Endocytic vesicles merge with lysosomes, where the pH is significantly lower than in serum, which leads to a decrease in the negative LO charge. Low pH values are optimal for protein hydrolysis by lysosomal enzymes. T1/2 of LO in serum was 1.23±0.10 h. Most of the antitumor enzymes that are injected i.v. have similar T1/2 in blood, for example, T1/2 of methionine gamma-lyase from C. tetani was 1.71±0.14 and L-asparaginase from E.coli about 3 h; for L-asparaginase from E. coli it was shown that human macrophages bind and degrade this enzyme [16-19]. While the value of LO T1/2 in blood is rather short, T1/2 of LO in the brain was 9.41±1.10 h. The longer persistence in the brain is possibly due to rather slow utilization of extracellular proteins in the brain»

Point 7: Lines 108-111 (page 4) and lines 179-181 (pages 6) are identical.  Please eliminate the repeat.

Response 7: We are sorry, we have corrected the mistake.

Point 8: Figure 5 (page 5). Up to 1 h time the dynamics of polyamine concentrations are as it may be expected. Put comes from Orn; Spd comes from Put and dcAdoMet; Spm comes from Spd and dcAdoMet. Respectively, the depletion of Orn pool must affect first Put and then Spd pool. This is typical for ornithine decarboxylase (ODC) inhibitors, most known is a-difluoromethyl ornithine (DFMO, Eflornithine®), which efficiently depletes Put and Spd pool, but is much less effective towards Spm one. However, after 24 h Spd concentration is not changed, while the concentration of Spm is about 70% and Put is of only 40% of the initial levels. In general, it is not easy to deplete Spm pool and does not change Spd concentration even in cell culture experiments. The observed dependences (Fig.5) are unusual and not discussed in the manuscript.

Response 8:  We have inserted into the "Discussion":

“Comparison of Fig. 3 and Fig. 5 shows that the drop in the concentration of L-orn and PA is not directly related. L-arg, along with L-orn, is a molecule from which PA can be synthesized through agmatine or after transformation into L-orn. PA homeostasis in tissue is achieved by a careful balance between synthesis, degradation and uptake. The first and what is often considered the rate-limiting step in the biosynthesis of the PA is catalyzed by the enzyme ornithine decarboxylase. Mammalian ornithine decarboxylase has a fast turnover, with a half-life as short as a few minutes, so its cellular level is thus rapidly changed when the rate of PA synthesis or degradation is changed [35,36]. It has been established that ornithine decarboxylase is inhibited by putrescin [37]. An initial drop in putrescine concentration can activate ornithine decarboxylase and cause an even greater drop in PA concentrations (Fig. 5). 24 h after LO introduction, the levels of nonessential amino acids L-orn (95.4±7.4 %) and L-arg (76.1±8.5 %) gradually move up to normal values, as well as the concentration of spermidine (Fig. 5). At the same time, concentrations of other PA remain significantly lower: spermine – 67.2±7.8 %, putrescine - 41.3±3.4 % of the initial. PA concentrations in the putrescine-spermidine-spermine sequence are not linearly related but are the result of complex processes. It has been established that enzymes for the synthesis and decomposition of PA are present in neurons, but absent in glial cells. However, after synthesis PA are transported by carriers into glial cells, where they are accumulated at higher concentrations than in neurons [38, 39]. Given the higher content of PA in glia and quantitative predominance of glial cells over neurons, it may be assumed that in our experiments with brain homogenates we determined the total PA amount, which reflects not only the process of PA synthesis and decomposition, but, as well the transfer and accumulation of PA in glia. Direct relationship between conversion of L-orn and L-arg to putrescine and then putrescine to spermidine and from spermidine to spermine and the decomposition of PA cannot be detected in experiments on a general homogenate of brain tissue.”

Point 9: lines 171-172 (page 6). Instead of “piperidine-2-carboxylate” it must be Δ1-piperideine-2-carboxylate

Response 9: We are sorry, we have corrected the mistake.

Point 10: line 185 (page 6). Why “… it can be assumed that the removal of L-glu, which is critical for the proliferation of tumor cells…” ? From Fig.4 it is clear that about 60% of Glu is left after 6 h. What do you mean under “removal” ? Besides, the centrality of Glu metabolism makes this amino acid crucial for normal cells too. Please modify the sentence. 

Response 10:  We modified the sentence and inserted a reference:

«The decrease in the L-glu concentration lasted as long as that of L-lys. L-glu is used in the organism for the biosynthesis of L-orn and L-arg. As these AA are partially destroyed by LO, it is possible that a drop in the L-glu concentration resulted from the effect of LO on L-orn and L-arg. Since LO shows antitumor effects in the body [12, 30], it can be assumed that the decrease of L-glu, which is critical for the proliferation of many tumor strains [31], may be one of the mechanisms of LO antitumor action along with L-lys depletion and the oxidative pathway [13]».

Point 11: lines 208-210 (page 6). “However, administration of these inhibitors is accompanied by a number of neurological side effects, which are apparently associated with the peculiarities of PA metabolism in the brain”. This is not a true statement. For example, DFMO (this drug was approved by WHO) is used for chemoprevention [Gerner E.W., et al. Biol.Chem.2018, 293, 18770–18778] and to cure late-stage of sleeping sickness disease [Steverding D. Parasites & Vectors2010, 3, 15]. The same is true also for some other inhibitors, see the review [Casero R.A., et al. “Polyamine metabolism and cancer: treatments, challenges and opportunities”. Nat.Rev.Cancer2018, 18, 681-695].

Response 11: We agree that there are different ornithine decarboxylase (ODC) inhibitors with different activities and different side-effects. As this work is not devoted to ODC we decided to omit the sentence: “However, administration of these inhibitors is accompanied by a number of neurological side effects, which are apparently associated with the peculiarities of PA metabolism in the brain”.

Point 12: line 215 (page 7). “… as the discovery…” Discovery seems too ambitious here. This is an interesting experimental fact, which still requires at least a proper discussion, but not a “discovery”.

Response 12: Thank you for indication of nonconformity. We substituted the sentence to: The decrease of PA level under the influence of LO can be proposed as a new mechanism of LO antitumor action.

Point 13: line 219 (page 7) and line 154 (page 5). References [15] and [23] are essential for the manuscript and ref [15] is placed even in conclusion. Both references are Abstracts. Please substitute these with the references on the papers in per-reviewed journals.

Response 13: We substituted reference [15].

We showed in Fig.1a that the dynamics of LO concentration changes after i.v. administration, determined by the method of enzyme immunoassay, practically coincides with the dynamics of enzymatic activity decrease, so we can conclude that enzyme immunoassay gives information about functionally active LO,  so we deleted reference [23].

Point 14: line 294 (page 8). “…followed by 20 μl benzoyl chloride (diluted in methanol 1:1)”. This sounds more than strange because of the formation of methyl benzoate in such a solution. Respectively, the excess of benzoyl chloride is unpredictable. This makes the results at least hardly reproducible, or might be even not completely reliable. As a rule, benzoyl chloride is used as such for pre-column modification of polyamines [Li B., et al. ACS Chem. Biol.2016, 11, 2782-2789], or it may be diluted with inert water-soluble organic solvents. For example, dansylation of polyamines is performed using a solution of dansyl chloride in acetone [Minocha R.& Long S. J. A 2004, 1035, 63–73].

Response 14: Our experiment was based on methods   Asotra S., Mladenov P. V., Burke R. D. Improved method for benzoyl chloride derivatization of polyamines for high-peformance liquid chromatography //Journal of Chromatography A. – 1987. – Т. 408. – С. 227-233. R. Sethi, S. Chava, S. Bashir and M. Castro, "An Improved High Performance Liquid Chromatographic Method for Identification and Quantization of Polyamines as Benzoylated Derivatives," American Journal of Analytical Chemistry, Vol. 2 No. 4, 2011, pp. 456-469. doi: 10.4236/ajac.2011.24055.

Point 15: Line 297 (page 8). For HPLC determination of polyamines 1,7-diaminoheptane is traditionally used as inner standard, but not “1,4-diamino heptane”. Did you use 1,4-diaminoheptane ?

Response 15: Thank you, it was misprinting, we used 1,7-diaminoheptane.

Point 16: It is known that the etiology of 20–30% of epithelial cancers is directly associated with inflammation. Spermine oxidase (SMOX) is a polyamine catabolic enzyme that is highly inducible by inflammatory stimuli resulting in the generation of hydrogen peroxide and acrolein-producing 3-aminopropanal; these lead to DNA damage and apoptosis [Sierra J.C., et al. Oncogene2020, 39, 4465–4474]. In the multiple intestinal neoplasia mouse model, treatment with SMOX inhibitor reduces enterotoxigenic Bacteroides fragilis induced colon tumorigenesis by 69% [Goodwin A.C., et al. Proc.Natl.Acad.Sci USA2011, 108, 15354–15359]. The same is true for gastric cancer [Chaturved R., et al. Gastroenterology2011, 141, 1696–1708]. Respectively, the use of hydrogen peroxide generating systems for treatment of cancer is comparatively delicate.

Response 16: We agree with the reviewer's comment on the caution of using hydrogen peroxide generating systems. Anticancer agents affect not only tumor cells, but also healthy cells. That is why the development of treatment regimens for tumor diseases is carried out under the control of tolerance and safety.

Reviewer 2 Report

The authors showed that fungal LO accumulates in the brain, where it catalyzes the degradation of Lysine, Arginine and Ornithine.

In Fig 3 the authors showed that, at 1h post LO injection, there is modest decrease of Orn concentration. In Fig 5 the authors demonstrated a significant reduction of PA content in the brain, starting at 0.17h post LO injection. The authors should explain why the reduction of putrescine (the first PA synthetized) starts at 0.17 h post LO injection and the concentration remains low up to 24h. The assumption that the degradation of Orn by fungal LO corresponds to a reduced synthesis of PA is not straightforward, considering also that putrescine is rapidly converted into spermidine. 

Author Response

Thank you very much for attention to our manuscript. It has been revised according to the reviewers’ suggestions.

Point 1: In Fig 3 the authors showed that, at 1h post LO injection, there is modest decrease of Orn concentration. In Fig 5 the authors demonstrated a significant reduction of PA content in the brain, starting at 0.17h post LO injection. The authors should explain why the reduction of putrescine (the first PA synthetized) starts at 0.17 h post LO injection and the concentration remains low up to 24h. The assumption that the degradation of Orn by fungal LO corresponds to a reduced synthesis of PA is not straightforward, considering also that putrescine is rapidly converted into spermidine. 

Response 1:  We thank the referee for the comment made. In line with the comments of the reviewers, we have expanded the discussion of the results of Fig. 3 and Fig. 5:

«Comparison of Fig. 3 and Fig. 5 shows that the drop in the concentration of L-orn and PA is not directly related. L-arg, along with L-orn, is a molecule from which PA can be synthesized through agmatine or after transformation into L-orn. PA homeostasis in tissue is achieved by a careful balance between synthesis, degradation and uptake. The first and what is often considered the rate-limiting step in the biosynthesis of the PA is catalyzed by the enzyme ornithine decarboxylase. Mammalian ornithine decarboxylase has a fast turnover, with a half-life as short as a few minutes, so its cellular level is thus rapidly changed when the rate of PA synthesis or degradation is changed [35,36]. It has been established that ornithine decarboxylase is inhibited by putrescin [37]. An initial drop in putrescine concentration can activate ornithine decarboxylase and cause an even greater drop in PA concentrations (Fig. 5). 24 h after LO introduction, the levels of nonessential amino acids L-orn (95.4±7.4 %) and L-arg (76.1±8.5 %) gradually move up to normal values, as well as the concentration of spermidine (Fig. 5). At the same time, concentrations of other PA remain significantly lower: spermine – 67.2±7.8 %, putrescine - 41.3±3.4 % of the initial. PA concentrations in the putrescine-spermidine-spermine sequence are not linearly related but are the result of complex processes. It has been established that enzymes for the synthesis and decomposition of PA are present in neurons, but absent in glial cells. However, after synthesis PA are transported by carriers into glial cells, where they are accumulated at higher concentrations than in neurons [38, 39]. Given the higher content of PA in glia and quantitative predominance of glial cells over neurons, it may be assumed that in our experiments with brain homogenates we determined the total PA amount, which reflects not only the process of PA synthesis and decomposition, but, as well the transfer and accumulation of PA in glia. Direct relationship between conversion of L-orn and L-arg to putrescine and then putrescine to spermidine and from spermidine to spermine and the decomposition of PA cannot be detected in experiments on a general homogenate of brain tissue».

Round 2

Reviewer 1 Report

Unfortunately, I have to restate that I cannot accept as chemically sound the benzoylation procedure that uses benzoyl chloride in methanol.

Benzoyl chloride reacts with methanol, the reaction is exothermic, and the resulting solution contains of methyl benzoate and an unknown amount of unreacted benzoyl chloride. The residual concentration of benzoyl chloride is determined by the conditions of solution preparation and also by the solution storage conditions and storage duration. The exact amount of benzoyl chloride in such a solution cannot be predicted. Therefore, at the very least, polyamine determination using this protocol is highly unlikely to yield reproducible results. Providing references for a protocol that contradicts basic chemistry knowledge does not eliminate the controversy. These references are either outdated or not convincing. New less controversial protocols describing the benzoylation of polyamines have been published and readily available.

Based on the above, determinations of the polyamine concentration will be good to repeat using a reliable benzoylation protocol. It may not be a problem if you have samples retained in the freezer.

In general, your study makes a good impression and may also be of interest to polyamine community.

In my opinion, the paper may be accepted for publication after the submission of newly obtained polyamine pool data.

Author Response

Dear Editor,

Thank you very much for attention to our manuscript.

We would like to clarify the adequacy of the used method for determining the PA.

Benzoyl chloride was mixed with methanol just prior to addition to the sample, so there was practically no time for reaction. To date, we have verified in a separate experiment that under the specified experimental conditions a violent reaction between benzoyl chloride and methanol does not occur, the mixture is not heated. 

We made a clarification of the volumes of reagents, which were used in the experiment. These refinements were made in connection with the fact that the beginning of the experiment was described for 1 g of tissue*

After adding a mixture of benzoyl chloride with methanol to the sample, both reagents were diluted 300 times. If we assume that the kinetics of the reaction between methanol and benzoyl chloride is of the second order, then the reaction would slow down by a factor of 9∙104 .

Before the determination of PA in the samples, calibration curves were constructed both in the presence and in the absence of methanol. They made it possible to adequately determine the concentration of standard samples of polyamines. The short-term mixing with methanol did not affect the shape of the calibration curves.

If the editor and the reviewer are not satisfied with our answer, we ask for additional time until November 12 to experimentally measure by NMR how significant is the formation of the product of the reaction between methanol and benzoyl chloride in these conditions.

__________________

*«PA levels in the brain were determined after converting them to benzoyl derivatives by high-performance liquid chromatography [43, 44]. 1,7-Diamino heptane was used as internal control. To precipitate the proteins, 2 ml HClO4 (1.0 M) was added to 0.2 g tissue. After centrifugation, 2.0 ml NaOH (2.0 M) was added to 1.0 ml of supernatant for neutralization.  Right before use (not longer than 1 min) benzoyl chloride (10 μl) was mixed with methanol (10 μl) and added to the mixture. After intensive stirring for 2 h, 1.0 ml of chloroform was added, and the sample was centrifuged for 10 min at 350 g. The chloroform fraction was removed, 1.0 ml of chloroform was re-added to the samples and samples were centrifuged. After that, the chloroform fractions were evaporated at 80 С˚. The dry residue was dissolved in 100 µl of 60% methanol. The composition of the mobile phase was methanol: water (60: 40), the volume of the samples injected to Lichrospher RP18 column was 20 µl and the chromatographic analysis time was 45 min. Substances were identified by absorption at 229 nm. The exit time from a chromatographic column for the benzoyl derivatives were: putrescine 12 min, spermidine 19 min, 1,7-diamino heptane 21 min and spermine 30 min».

We made additional experiments and analyzed the literature data on reaction of benzoyl chlorides with amines

  • We have found (by NMR 1H) that when equal volumes of BzCl and MeOH were mixed (50 μl each, mol ratio approx. 1:3), no exothermic reaction was observed. After 10 min at r. t., this mixture was diluted it with CDCl3 for NMR analysis. Ratio of BzCl and BzOMe was 45:55, i.e. reaction of pure BzCl with pure MeOH is relatively slow. In our experiment in the manuscript we did not wait for 10 min.
  • Analysis of literature data on alcoholysis, aminolysis and hydrolysis of acetyl (and benzoyl) chlorides show that nucleophilic substitution reaction for BzCl proceeds two orders of magnitude slower in comparison with AcCl. And the most important is that water/methanol/BzCl mixture is an applicable media for benzoylation of m-nitroanilin, which is less nucleophilic in comparison with aliphatic polyamines. And even for such bad nucleophile rate, the constant for aminolysis of BzCl is "1000-2000 times greater than the first order methanolysis rate constant from methanol to 20% methanol/water" [Bentley, T. William, Gareth Llewellyn, and J. Anthony McAlister. "SN2 mechanism for alcoholysis, aminolysis, and hydrolysis of acetyl chloride." The Journal of organic chemistry22 (1996): 7927-7932].
  • A similar scheme for the analysis of mixtures of polyamines, which included dissolving of benzoyl chloride in methanol was also proposed earlier [Alsouz M. A. Separation and Determination of Benzoylated Polyamines (Spermidine and Spermine) using HPLC Techniqe.https://core.ac.uk/download/pdf/234666577.pdf]

In this work, as in ours, a high convergence of results was shown.

Therefore, the method used in our work is not compromised by the reaction between BzCl and MeOH, has been used in other publications and gives sufficiently precise results to justify conclusions of the manuscript.  We ask the reviewer and editor to consider our arguments when judging the possibility of publishing the manuscript.

Round 3

Reviewer 1 Report

The paper may be accepted for publication.